# Towards Open-vocabulary HOI Detection with Calibrated Vision-language Models and Locality-aware Queries

## ABSTRACT

The open-vocabulary human-object interaction (Ov-HOI) detection aims to identify both base and novel categories of human-object interactions while only base categories are available during training. Existing Ov-HOI methods commonly leverage knowledge distilled from CLIP to extend their ability to detect previously unseen interaction categories. However, our empirical observations indicate that the inherent noise present in CLIP has a detrimental effect on HOI prediction. Moreover, the absence of novel human-object position distributions often leads to overfitting on the base categories within their learned queries. To address these issues, we propose a two-step framework named, **CaM-LQ**, **Ca**librating visual-language **M**odels, (e.g., CLIP) for open-vocabulary HOI detection with **L**ocality-aware **Q**ueries. By injecting the fine-grained HOI supervision from the calibrated CLIP into the HOI decoder, our model can achieve the goal of predicting novel interactions. Extensive experimental results demonstrate that our approach performs well in open-vocabulary human-object interaction detection, surpassing state-of-the-art methods across multiple metrics on mainstream datasets and showing superior open-vocabulary HOI detection performance, e.g., with 4.54 points improvement on the HICO-DET dataset over the SoTA CLIP4HOI on the UV task with the same backbone ResNet-50. Our codes are available at: https://anonymous.4open.science/r/cam_lq.

## KEYWORDS

Human-object Interaction Detection, Open-vocabulary Learning, Vision-Language Models

### ACM Reference Format:

Anonymous Author(s). 2018. Towards Open-vocabulary HOI Detection with Calibrated Vision-language Models and Locality-aware Queries. In *Proceedings of Make sure to enter the correct conference title from your rights confirmation emai (Conference acronym 'XX).* ACM, New York, NY, USA, 9 pages. https://doi.org/XXXXXXX.XXXXXXX

## 1 INTRODUCTION

Human-object Interaction (HOI) detection, a fundamental task of human-centered scene understanding, has gained considerable attention across diverse domains, such as Image Captioning [27], Visual Question Answering (VQA) [33], and video analysis [5]. Notably, prevailing methods [28, 40] have demonstrated stunning

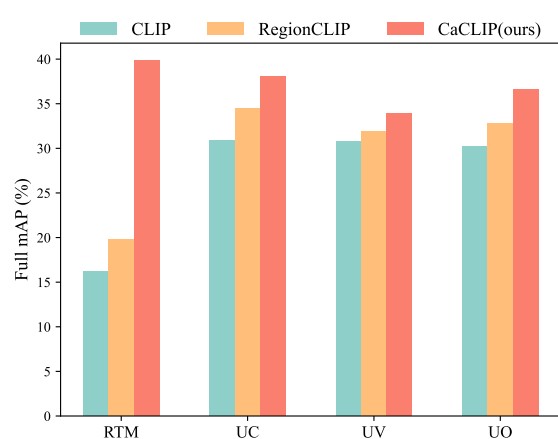

**Figure 1: Noise of V&L mode comparison. We investigate three different V&L models on RTM, UC, UV, UO tasks, which denotes the region text matching and HOI detection under settings where compositions, verbs, and objects are unseen, respectively. CaCLIP denotes our calibrated CLIP.**

performance within the confines of closed-set scenarios. Nevertheless, the inherent complexity of real-world interactions between humans and objects gives rise to a multitude of interaction categories that are not comprehensively encapsulated by existing HOI datasets, such as V-COCO [8] and HICO-DET [2]. Hence, there is a compelling necessity to shift attention toward the exploration of Open-vocabulary Human-object Interaction (Ov-HOI) detection.

With the widespread adoption of large-scale pretrained vision-language models (VLMs), such as CLIP [29], prior works [20, 32, 34] have showcased the impressive generalization capabilities of VLMs in open-vocabulary scenarios [7, 42]. However, these models encounter three primary limitations: (1) struggling in fine-grained HOI detection, especially for local regions, due to their adoption of a global image-level knowledge distillation strategy [20, 32] to guide their image encoders towards learning CLIP-like embeddings; (2) prone to overfitting on the spatial features of base categories, making them incapable of handling scenarios with substantial spatial distribution disparities between seen and unseen HOI categories; (3) misalignment of HOI visual and language knowledge, attributed to intrinsic noise introduced by pretrained multimodal models like CLIP. To empirically assess these limitations, Figure 1 illustrates the performance evaluation of three multimodal features: CLIP, RegionCLIP [42], and our calibrated CLIP, in HOI classification tasks under various scenarios. The former two features exhibit obviously inferior performance compared to our calibrated CLIP. For more analysis, please refer to Sec. 5.4.

To overcome the aforementioned issues, we propose a novel model, denoted as CaM-LQ, aimed at calibrating visual language

models with fine-grained HOI priors and then training an open-vocabulary HOI detector with locality-aware queries. More concretely, we develop a two-step training mechanism: (1) calibrating CLIP with HOI priors, and (2) performing open-vocabulary HOI detection. In the first step, we suppress the intrinsic noise in visual language models, such as CLIP, by calibrating CLIP features with HOI knowledge through training two parallel adapters [6]. This calibration ensures that the learned visual-semantic space is equipped with corrective HOI concepts, thereby enhancing its compatibility for HOI detection.

Moving on to the second step, we initially employ a pre-trained object detector, e.g., DETR [1], to identify objects and construct pairwise queries for feasible human-object pairs. Specifically, we utilize spatial embeddings of human-object pairs to refine human and object visual features and employ human-object queries to decode the global image feature with spatial priors. Subsequently, for each detected pair, we encode union box embeddings and compute their similarities with pre-defined text embeddings using our calibrated pre-trained model at the first stage. These similarities serve as fine-grained supervision to guide the training of the open-vocabulary HOI detection network through a logit distillation mechanism.

In summary, our contributions can be outlined as follows:

- Through comprehensive empirical experiments of V&L models, we observe the presence of significant intrinsic noise in their embeddings, which proves to be detrimental to open-vocabulary HOI detection.
- We devise an approach to calibrate CLIP with HOI priors, coupled with a fine-grained logit distillation strategy, to alleviate the impact of intrinsic noise in the embeddings.
- We propose to inject spatial priors into HOI queries to decode pairwise HOI features, which will help the mode focus on the interaction point and capture a nuanced relationship.
- Our model surpasses state-of-the-art approaches with a large margin on many metrics of Ov-HOI detection, e.g., surpassing the SoTA CLIP4HOI with 4.54 mAP on the HICO-DET dataset on the UV task with the backbone ResNet-50.

## 2 RELATED WORK

In this section, we will carefully review literature from the aspects of HOI detection, open-vocabulary HOI detection and HOI detection with CLIP.

### 2.1 HOI Detection

The current mainstream HOI detection methods can be categorized into two paradigms: one-stage and two-stage. One-stage approaches [14, 17, 30] treat HOI detection as a set prediction problem, adopting parallel prediction methods to locate objects and predict interaction categories. This requires a decoder to extract effective interaction information from the learned query. However, these methods are often constrained by excessive noise in the learned queries, leading to an inability to decode the correct interaction categories. Additionally, as the query learns from the training set, it may result in over-fitting to the position distribution of human and base object [25]. On the other hand, two-stage methods [18, 28] often utilize a pre-trained detector to locate objects and predict categories. Therefore, they can leverage off-the-shelf features to construct the query,

imbuing it with sufficient interaction information. Our approach adopts a two-stage step, based on a pre-trained detector [1], to enhance interaction prediction capabilities.

### 2.2 Open-vocabulary HOI Detection

Open-vocabulary HOI detection [10, 38] aims to predict base and novel HOI categories with only base HOI labels during training. Due to the complexity of the composition of human and object interaction, it is almost impossible to construct all HOI categories. Therefore, open-vocabulary HOI detection can be more effectively applied in real-world scenarios. The task currently has three main scenarios: Unseen Composition(UC), Unseen Object(UO), Unseen Verb(UV). Recently, ConsNet [23] employed explicit and implicit knowledge of HOI from consistency graphs and word embeddings. However, these methods are constrained by the limitations of HOI datasets and cannot fully explore the capability of open-vocabulary HOI detection. In light of this, recent methods [20, 25] extract knowledge from vision-language pre-trained model [29] to improve open-vocabulary performance and have achieved promising scores.

### 2.3 HOI Detection with CLIP

Recently, CLIP has successfully implemented contrastive learning with large-scale image-text pair data gathered from the internet and achieved powerful zero-shot performance. The approaches to leverage CLIP knowledge in HOI tasks can be divided into two groups. The first group utilizes CLIP to extract global features. For example, GEN-VLKT [20] encodes the entire image with CLIP and applies supervised learning on the global image features to the image encoder in the network. HOICLIP [26], integrates CLIP global embeddings with backbone features, followed by a decoder to predict interaction categories. CLIP4HOI [25] employs the global feature from CLIP to introduce prior knowledge to aid the final prediction of interaction categories. The other group, however, involves utilizing CLIP to extract region features. For instance, EoID [34] proposed to inject more regional supervision information into the predictions by union box features. Similarly, we borrow the idea of EoID to inject fine-grained supervision into the decoder.

However, none of the aforementioned methods consider the noise brought by CLIP which is trained on large-scale image-caption datasets, leading the bias towards the object category, rather than HOI category. It is necessary to calibrate CLIP features with HOI priors and eliminate noise. In light of this, we propose a pre-training step to calibrate CLIP with HOI priors, enabling a visual semantic space with rich HOI priors.

## 3 PROBLEM STATEMENT

In this section, we introduce the setting of HOI detection in both fully supervised and open-vocabulary settings. Denote $O = \{o_1, o_2, ..., o_{N_o}\}$ be the object category, $\mathcal{V} = \{v_1, v_2, ..., v_{N_v}\}$ be the interaction verbs and $C$ denote all the feasible composition of verb-object pairs, i.e., $C = \{(v_i, o_j)|v_i \in \mathcal{V}; o_j \in O\}$. Let $I$ denote an input image, with corresponding labels $T = \{\mathcal{B}, \mathcal{Y}\}$ where $\mathcal{B}$ is a set of bounding boxes including human bounding boxes $\mathcal{B}_h$ and object bounding boxes $\mathcal{B}_o$, and $\mathcal{Y}$ denote a set of HOI triplets. Each $\langle b_h, b_o, c_i \rangle$ in $\mathcal{Y}$ is a HOI triplet, where $c_i \in C$.

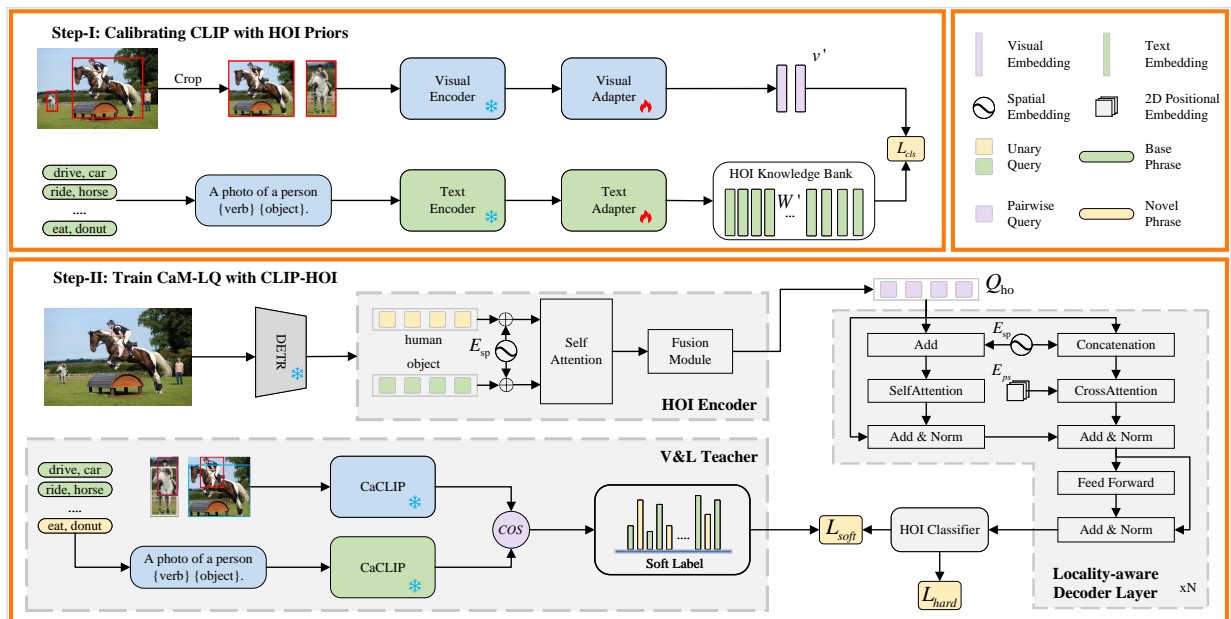

**Figure 2: Our proposed open-vocabulary HOI model CaM-LQ. Top: the pre-training procedure for fine-tuning CLIP with HOI priors to get CaCLIP with rich HOI knowledge. Bottom: our two-stage open-vocabulary HOI detector. Firstly, we detect all humans and objects with a pre-trained object detector. Then we encode pairwise queries with visual features and spatial priors. Finally, the pairwise queries are fed into the decoder and further refined in each layer with spatial embeddings.**

In a fully supervised scenario, all the feasible verb-object pairs $\langle v_i, o_i \rangle$ are included in the training set. However, for open-vocabulary settings, only base verb-object pairs are seen during training and the model needs to process novel pairs during testing. Let $V_{base} \subset \mathcal{V}$, $O_{base} \subset O$ and $C_{base} \subset C \setminus C_{novel}$. According to whether verbs and objects are available during training, there are three different settings: (1) Unseen Object (UO), where for all $\langle v_i, o_i \rangle \in C_{novel}$, we have $v_i \in \mathcal{V}_{base}$ and $o_j \in O_{novel}$ (2) Unseen verb (UV), where for all $\langle v_i, o_i \rangle \in C_{novel}$, we have $v_i \in \mathcal{V}_{novel}$ and $o_j \in O_{base}$ (3) Unseen Composition (UC), where for all $\langle v_i, o_i \rangle \in C$, we have $v_i \in \mathcal{V}_{base}$ and $o_j \in O_{base}$.

## 4 METHOD

In this section, we introduce our proposed **CaM-LQ**, **Ca**librating visual-language **M**odels, and training an open-vocabulary HOI detector with **L**ocality-aware **Q**ueries for open-vocabulary HOI detection. As illustrated in Figure 2, our CaM-LQ employs a two-step training framework, encompassing four primary components: the teacher model CaCLIP, a pre-trained object detector, an HOI encoder, and a pairwise interaction decoder. Firstly, we conduct fine-tuning of the CLIP model, incorporating HOI priors to imbue HOI-specific knowledge and yielding a specialized V&L teacher model denoted as CaCLIP. The HOI transformer-based encoder is employed, utilizing query embeddings and box coordinates as inputs, integrating visual and spatial features to construct HOI queries. Subsequently, the interaction decoder integrates these queries with box spatial priors, and backbone features serving as key-value pairs, to predict the action in each HOI triplet. Ultimately, we compute cosine similarities between visual features and HOI text embeddings.

These similarity scores are utilized as soft labels to guide the action prediction, facilitating knowledge transfer to the HOI decoder.

### 4.1 Calibrating CLIP with HOI priors

Our empirical observations (seeing Sec.5.4 for more detail.) indicate that the raw CLIP feature can not handle HOI prediction well, primarily due to intrinsic noise stemming from the training on extensive web image-caption pairs. To address this limitation, we propose a calibration strategy to imbue CLIP with HOI-specific knowledge and align HOI visual and text information at a finer granularity, specifically at the region level.

However, due to the limited data availability, direct fine-tuning of all parameters of CLIP proves challenging. Inspired by [6], we adopt a more lightweight fine-tuning strategy by introducing two adapters, for the image and text encoders. In default, our adapters are implemented by MLPs and are positioned subsequent to the original CLIP image and text encoders. In practice, we also test other alternative adapter architectures, and for more details, please refer to the supplementary materials.

We employ human-annotated HOI labels to calculate the union box coordinates for each human-object pair, i.e., the minimum bounding rectangle. These union boxes are utilized to crop the image as:

$$\mathcal{I}_{crop} = \Phi_{crop}(\mathcal{I}, \Phi_{union}(b_h, b_o)) \tag{1}$$

$$v'_i = \text{Adapter}_{img}(\text{CLIP}_{img}(\mathcal{I}_{crop})) \tag{2}$$

Where $v'_i$ denotes the embedding of the union region, $\Phi_{union}$ represents the operation computing the union box for the human-object pair, $\Phi_{crop}$ signifies the operation of cropping the bounding box

region from the image, $\text{CLIP}_{img}$ is the function responsible for encoding the image using the CLIP visual encoder and $\text{Adapter}_{img}$ denotes the image adapter.

As for the text branch, we utilize CLIP to encode a predefined set of HOI texts, by a template, e.g., "a photo of a person verb object," where "verb" and "object" represent labels in an HOI triplet. Subsequently, we establish an HOI knowledge bank denoted as $W = [w_1, w_2, ..., w_{|C|}]$, which serves as a classifier. By feeding text features into the HOI adapter, we derive a refined HOI knowledge bank $W'$. The cosine similarity between the obtained human-object region feature and predefined text features is then computed:

$$\hat{s} = \text{Softmax}([sim(v'_i, w'_1), ..., sim(v'_i, w'_{|C|})]) \qquad (3)$$

Where $sim(v'_i, w'_j) = (v'_i \cdot w'_i)/(||v'_i|| \cdot ||w'_i||)$. Finally, a binary cross-entropy loss is employed for multi-label classification, aligning pairwise region features with corresponding HOI texts. Following this calibration process, we acquire a calibrated CLIP endowed with HOI priors, dubbed CaCLIP, which attains proficiency in a specialized HOI visual-semantic space, rendering it amenable for subsequent applications in open-vocabulary HOI detection.

## 4.2 Visual and Spatial-priors Encoding

In this section, we present how we encode visual features and inject spatial priors into the HOI queries. Following [39], we adopt a two-stage paradigm for HOI detection. Specifically, we first input an image into DETR [1] and obtain the global visual features and a set of region proposals. As [39], we post-process the region proposals with non-maximum suppression (NMS) and filter the DETR queries with confidence scores lower than a threshold. The post-processed results are denoted as $(\hat{B}, \hat{S}, \hat{O}, H)$, where $\hat{B} \in R^{N_{pred} \times 4}$, $\hat{S} \in [0, 1]^{N_{pred}}$, $\hat{O} \in \{0, 1, ..., |O| - 1\}^{N_{pred}}$ and $H \in R^{N_{pred} \times C_d}$ refer to predicted boxes, confidence scores, object categories and query features respectively, where $N_{pred}$ denotes the number of queries after filtering.

Since the queries obtained from a pre-trained object detector [1] are designed for localization and classification, directly employing them as interaction queries can not obtain ideal results, due to the fact that they typically focus on object-level information rather than the pairwise interactive concepts. Hence, to further adapt them to represent HOI visual embeddings, we employ a self-attention mechanism to refine them. The objective of self-attention mechanism is to enable human and object queries to focus on each other's features and complement each other.

Concretely, we compute the sinusoidal embeddings [31] of box center $c_x$, $c_y$, width $c_w$ and height $c_h$, by $\Theta : R \to R^d$

$$\Theta(x)_{2i} = sin(\frac{x}{\tau^{2i/d}}), \Theta(x)_{2i-1} = cos(\frac{x}{\tau^{2i/d}}) \qquad (4)$$

where $d$ is half of the $C_d$, $i = 1, 2, ..., d/2$ and $\tau$ is a temperature parameter.

With the coordinates embeddings, we concatenate them to form the spatial embeddings $E_{sp} \in R^{N_{pred} \times 2C_d}$ and the object features are refined through self-attention layers as follows:

$$H' = \text{Self\_Attn}(H, \text{Proj}(E_{sp})) \qquad (5)$$

where $H' \in R^{N_{pred} \times C_d}$, Self Attn and Proj denote box spatial embeddings, refined object features, self-attention layers and projection layers respectively.

Subsequently, we keep all viable pairs, i.e., excluding those where the subject category is not human. Thus, human-object queries $Q_{ho} \in R^{N_{pair} \times C_r}$ for feasible pairs are constructed through concatenation and a two-layers MLP to fuse human and object features by the following operations:

$$Q_{ho} = \text{MLP}(\text{CAT}(H'_{\text{human}}, H'_{\text{object}})) \qquad (6)$$

where $H'_{\text{human}}$ and $H'_{\text{object}}$ represent the human and object features of feasible pairs in the spatially refined features $H'$. So far, $Q_{ho} \in R^{N_{pair} \times C_r}$ is enriched with detailed spatial information, which can facilitate the HOI decoder to pay more attention to the objects close to a human.

## 4.3 Locality-aware Interaction Decoder

In this section, we seek to use refined interaction queries to predict HOIs from visual features via a locality-aware interaction decoder. Following [40], we apply the backbone ResNet C5 [9] features in default and further refine it with lightweight window attention layers [24], where the refined features are denoted as $F' \in R^{h \times w \times C_d}$.

One-stage HOI approaches [20, 26] widely incorporate learnable position embeddings to ensure that the initialized queries capture interaction pairs at different locations. However, for a two-stage counterpart, since objects have already been localized and form human-object pairs, adding learnable vectors to capture position information becomes less meaningful. Inspired by this, we argue that the localization information of humans and objects helps the decoder in learning the interaction between human-object pairs. Hence, we concatenate the sinusoidal spatial information of humans and objects in feasible pairs to obtain pairwise spatial embeddings $E'_{sp} \in R^{N_{pair} \times 4C_d}$. Each pairwise spatial embedding $e'_{sp}$ is calculated as follows:

$$e'_{sp} = \text{CAT}(e_{\text{human}}, e_{\text{object}}) \qquad (7)$$

where $e_{\text{human}}, e_{\text{object}} \in E_{sp}$ denote spatial embedding of human and object in each feasible pair.

Subsequently, we develop a spatial prior enhancement module in each decoder layer by the sinusoidal embeddings (Eq.4) again. Concretely, we inject the box spatial priors $E'_{sp}$ from Eq.(7) and sinusoidal positional embeddings $E'_{ps}$ into each decoder layer to augment the query $Q_{ho}$ and image features $\mathcal{F}'$ respectively. Taking the query in the $l$-th layer as an example, the augmentation operation is calculated as follows:

$$\begin{aligned} Q'_l &= \text{LN}_1(\text{MHSA}(Q_l + E'_{sp}) + Q_l) \\ Q''_l &= \text{MHCA}(\text{CAT}(Q_l, E'_{sp}), \text{CAT}(F', E'_{ps}), F') \\ Q'''_l &= \text{LN}_2(Q'_l + Q''_l) \\ Q_{l+1} &= \text{LN}_3(Q'''_l + FFN(Q'''_l)) \end{aligned} \qquad (8)$$

where LN, MHSA, MHCA, CAT, FFN denotes layer normalization, multi-head self-attention, multi-head cross attention, concatenation and feed-forward network, respectively. Projection layers and dropout operations are omitted for simplicity. It is worth noting that the first layer $l = 0$, $Q_0$ is set to $Q_{ho}$ from Eq.(6). Finally, the output

hidden state embedding of the last layer, i.e., $Q_N$ should incorporates interactive feature information of human-object pairs and is employed to predict interactions via a binary classifier, yielding $S_v \in \mathbb{R}^{N_{pair} \times |\mathcal{V}|}$.

## 4.4  Calibrated CLIP for Ov-HOI Detection

Owing to the unavailability of novel categories during training, we turn to CaCLIP, serving as a teacher model, to exploit its open-vocabulary HOI knowledge. With the huge discrepancy of the features between the teacher and the HOI decoder, we adopt the logits from CaCLIP as the guidance knowledge instead of its V&L features, to avoid the noise feature in the V&L teacher. Note that though EoID [34] adopts the similar strategy to inject CLIP signals as supervisory information, the significant noise presented in CLIP severely impedes the accurate alignment between image regions and their corresponding HOI texts. This results in the supervision scores providing less satisfactory guidance than ours provided by CaCLIP.

In implementation, we use the union box of each feasible pair to crop the input image and compute visual embedding through CaCLIP. Afterward, the cosine similarities with the predefined HOI texts, i.e., (verb, object), are normalized by a softmax operation. Subsequently, those similarity scores are mapped to action logits $S_{clip} \in \mathbb{R}^{N_{pair} \times |\mathcal{V}|}$ where invalid actions are masked out with the priors provided from DETR. Then, we treat the logits $S_{clip}$ from CaCLIP as a soft label to guide the interaction prediction. Note that $S_{clip}$ only influences categories not covered by the ground truth. We conduct a mask operation to choose the CLIP logit for computing losses:

$$\text{SoftMask}_{(i,j)} = \begin{cases} 1 & \text{if } S_{clip(i,j)} \neq 0 \text{ and } S_{gt(i,j)} \neq 1 \\ 0 & \text{otherwise} \end{cases} \quad (9)$$

where $S_{gt} \in [0,1]^{N_{pair} \times |\mathcal{V}|}$ denote ground truth labels of action categories. In practice, we use $\text{SoftMask}_{(i,j)}$ to multiply losses calculated by the soft CLIP label.

## 4.5  Training and Inference

**Training:** Given action logits, the total loss comes from two parts: the hard loss with ground truth labels and the soft loss with CLIP supervision labels. We apply focal loss [21] to mitigate the impact of imbalanced data distribution and hyper-parameter $\lambda$ to balance the two terms:

$$\mathcal{L}_{total} = \mathcal{L}_{hard} + \lambda \mathcal{L}_{soft} \quad (10)$$

**Inference:** We combine the confidence scores $(S^h, S^o)$ from DETR as prior knowledge to obtain the final prediction scores following [40]:

$$S_p = (S_h S_o)^{(1-\varphi)} S_v^{\varphi} \quad (11)$$

where $S_p \in N^{pair \times |\mathcal{V}|}$ and $\varphi \in [0,1]$ is a hyperparameter and $S_v$ is the predicted action score by the HOI decoder.

## 5  EXPERIMENTS

In this section, we will carefully elaborate our comprehensive experimental evaluation on three benchmark datasets. More results can be found in the supplementary materials.

## 5.1  Experimental setup

**Datasets:** we conduct downstream experiments on two popular benchmarks: HICO-DET [2] and V-COCO [8]. Specifically, HICO-DET comprises 37,633 training images, 9,546 testing images, with 80 object categories, 117 action categories, and a total of 600 interaction relationship combinations. V-COCO consists of 2,533 training images, 2,876 validation images, 4,946 testing images, featuring 80 object categories and 26 action categories.

**Metrics:** We use mean average precision (mAP) as the evaluation metric. For a feasible pair, we consider it as a positive sample only if the Intersection over Union (IOU) between the bounding boxes of the person and object and the ground truth label exceeds the threshold and the corresponding label is assigned to positive sample. Otherwise, it is treated as a negative sample.

**Open-vocabulary Setups:** As described in Section 3, we conducted experiments based on the UO, UV, and UC settings. Additionally, the UC setting is extended to RF-UC, where tail HOI categories are selected as the novel, and NF-UC setting, which contains head HOI categories as the novel. For HICO-DET, the division between base and novel classes for all the Ov-HOI tasks follows the EoID [34] protocol. As for V-COCO, we are the first to report the UV results, please refer to the appendix for more details.

## 5.2  Implementations

We use the fine-tuned DETR with the backbone ResNet-50 [9] as our small model CaM-LQ$_s$ in default and a large counterpart CaM-LQ$_l$ with more powerful object detector DETR [15] of Swin-L [24] backbone. Following [39], we adopt the same filtering scheme only keeping the detections with a confidence score higher than 0.2. Besides, the distillation is built upon the CLIP, which takes ViT-B/16 [4] as its backbone. The CLIP adapters are instantiated as a two-layer MLP for both image and text branch. The hyper-parameter $\lambda$ and $\varphi$ are set to 400 and 0.26 respectively. For the selection of $\tau$, we follow [22] and set it to 20 to make the sinusoidal encoding in the Transformer more suitable for vision tasks. We use $\alpha = 0.5$ and $\gamma = 0.1$ in our focal loss, the same as previous work [40]. The object hidden features from DETR have a dimension of $C_d$ of 256, and the query features of the decoder are set to $C_r$ of 384. For more details, please refer to the appendix.

## 5.3  Comparison with the state-of-the-arts

*5.3.1  Effectiveness for Open-vocabulary HOI detection.* In Table 2, we report the results on HICO-DET under five different settings as [25]: UC (Unseen Composition), RF-UC (Rare First Unseen Combination), NF-UC (Non-rare First Unseen Combination), UO (Unseen Object), UV (Unseen Verb). On the UC setting, our approach achieves SoTA for small version and even gain an mAP gain of 10.05, 10.84 and 10.89 for unseen, seen, and full results with a larger backbone. As for the RF-UC setting, we demonstrate the robust generalization capability of our approach to achieve a remarkable improvement of 3.82 and 6.99 points on unseen categories with different backbones. When it turns to the NF-UC setting, our CaM-LQ can also outperform all the previous models. Besides, our model also shows considerable improvements on the UO and UV, confirming its strong generalizability to novel objects and actions.

| Method | Backbone | Type | | | | | | | | | | | |
| --- | --- | --- | --- | --- | --- | --- | --- | --- | --- | --- | --- | --- | --- |
| | | UC | | | RF-UC | | | NF-UC | | | UO | | |
| | | Unseen | Seen | Full | Unseen | Seen | Full | Unseen | Seen | Full | Unseen | Seen | Full |
| *One-stage methods* | | | | | | | | | | | | | |
| ATL [12] | R50 | - | - | - | 9.18 | 24.67 | 21.57 | 18.25 | 18.78 | 18.67 | 15.11 | 21.54 | 20.47 |
| GEN-VLKT† [20] | R101+ViT-32 | 21.36 | 32.91 | 30.56 | 21.36 | 32.91 | 30.56 | 25.05 | 23.38 | 23.71 | 10.51 | 28.92 | 25.63 |
| CDT [43] | R50 | 18.06 | 23.34 | 20.72 | - | - | - | - | - | - | - | - | - |
| EoID [34] | R50+R50 | 23.01 | 30.39 | 28.91 | 22.04 | 31.39 | 29.52 | 26.77 | 26.66 | 26.69 | 26.77 | 26.66 | 26.69 |
| HOICLIP [26] | R50+ViT-16 | 23.15 | 31.65 | 29.93 | 25.53 | 34.85 | 32.99 | 26.39 | 28.10 | 27.75 | 16.20 | 30.99 | 28.53 |
| DiffHOI-S [36] | R50 | - | - | - | 24.13 | 32.93 | 31.08 | 26.57 | 25.55 | 25.75 | 9.42 | 29.79 | 26.22 |
| LOGICHOI [19] | R50 | 25.97 | 34.93 | 33.17 | 25.97 | 34.93 | 33.17 | 26.84 | 27.86 | 27.95 | 26.84 | 27.86 | 27.95 |
| RLIPv2 [37] | R50 | - | - | - | 21.45 | 35.85 | 32.97 | 22.81 | 29.52 | 28.18 | - | - | - |
| *Two-stage methods* | | | | | | | | | | | | | |
| VCL [11] | R50 | - | - | - | 10.06 | 24.28 | 21.43 | 16.22 | 18.52 | 18.06 | - | - | - |
| FCL [13] | R50 | - | - | - | 13.16 | 24.23 | 22.01 | 18.66 | 19.55 | 19.37 | 15.54 | 20.74 | 19.87 |
| ConsNet [23] | R50 | 16.99 | 20.51 | 19.81 | - | - | - | - | - | - | 19.27 | 20.99 | 20.71 |
| ADA-CM [18] | R50+ViT-16 | - | - | - | 27.63 | 34.35 | 33.01 | 32.41 | 31.13 | 31.39 | - | - | - |
| OpenCat* [41] | R101+ViT-16 | - | - | - | 21.46 | 33.86 | 31.38 | 23.25 | 28.04 | 27.08 | 23.84 | 28.49 | 27.72 |
| CLIP4HOI [25] | R50+ViT-16 | 27.71 | 33.25 | 32.11 | 28.47 | 35.48 | 34.08 | 31.79 | 28.26 | 28.90 | 27.71 | 33.25 | 32.11 |
| DHD [35] | R101 | - | - | - | 23.32 | 30.09 | 28.53 | 27.35 | 22.09 | 23.14 | 27.05 | 27.87 | 27.73 |
| *CaM-LQ$_s$ (ours)* | R50+ViT-16 | **29.93** | **35.84** | **34.66** | **32.29** | **36.57** | **35.59** | **36.92** | **31.22** | **31.56** | **31.44** | 32.73 | **32.58** |
| QAHOI† [3] | Swin-L | 21.93 | 27.84 | 26.66 | 19.35 | 29.37 | 27.06 | 28.28 | 20.19 | 21.81 | 23.54 | 28.74 | 27.87 |
| PViC† [40] | Swin-L | 23.34 | 28.06 | 27.11 | 19.47 | 29.47 | 27.17 | 32.08 | 25.50 | 26.82 | 24.11 | 30.54 | 29.46 |
| DiffHOI-L [36] | Swin-L | - | - | - | 28.76 | 38.01 | 36.16 | 29.45 | 31.68 | 31.24 | 5.75 | 35.08 | 30.11 |
| *CaM-LQ$_l$ (ours)* | Swin-L+ViT-16 | **34.39** | **38.90** | **38.00** | **35.75** | **41.06** | **36.81** | **42.22** | **31.81** | **33.89** | **36.36** | **36.65** | **36.61** |

Table 1: Open-vocabulary results on HICO-DET. UC, UO denote unseen composition and unseen object settings. RF-UC and NF-UC refer to rare-first UC and non-rare-first UC respectively. The † denotes our implementation based on their released code.

| Method | Backbone | Type | Unseen | Seen | Full |
| --- | --- | --- | --- | --- | --- |
| *One-stage methods* | | | | | |
| GEN-VLKT [20] | R101+ViT-32 | UV | 20.96 | 30.23 | 28.74 |
| CDT [43] | R50 | UV | 19.68 | 21.45 | 15.17 |
| EoID [34] | R50+R50 | UV | 22.71 | 30.73 | 29.61 |
| HOICLIP [26] | R50+ViT-32 | UV | 24.30 | 32.19 | 31.09 |
| DiffHOI-S [36] | R50 | UV | 23.10 | 30.91 | 29.72 |
| LOGICHOI [19] | R50 | UV | 24.57 | 31.88 | 30.77 |
| *Two-stage methods* | | | | | |
| ConsNet [23] | R50 | UV | 14.12 | 20.02 | 19.04 |
| OpenCat* [41] | R101+ViT-16 | UV | 19.48 | 29.02 | 27.43 |
| CLIP4HOI [25] | R50+ViT-16 | UV | 26.02 | 31.14 | 30.42 |
| DHD [35] | R101 | UV | 17.92 | 28.13 | 26.43 |
| *CaM-LQ$_s$ (ours)* | R50+ViT-16 | UV | **30.56** | **32.27** | **31.98** |
| QA-HOI† [3] | Swin-L | UV | 10.64 | 28.19 | 25.26 |
| PViC† [40] | Swin-L | UV | 12.18 | 28.03 | 25.39 |
| DiffHOI-L [36] | Swin-L | UV | 24.20 | 36.81 | 35.04 |
| *CaM-LQ$_l$ (ours)* | Swin-L+ViT-16 | UV | **31.16** | **36.89** | **36.16** |

Table 2: Open-vocabulary results on HICO-DET under the unseen verb (UV) setting. The † denotes our implementation based on their released code.

| Method | Backbone | HICO-DET | | | V-COCO | |
| --- | --- | --- | --- | --- | --- | --- |
| | | Full | Rare | Non-Rare | $AP^{S1}_{role}$ | $AP^{S2}_{role}$ |
| GEN-VLKT [20] | R101+ViT-32 | 34.95 | 31.18 | 36.08 | 63.6 | 65.9 |
| HOICLIP [26] | R50+ViT-32 | 34.69 | 31.12 | 35.74 | 63.5 | 64.8 |
| LOGICHOI [19] | R50 | 35.47 | 32.03 | 36.22 | 64.4 | 65.6 |
| DiffHOI-S [36] | R50 | 34.41 | 31.07 | 35.40 | 61.1 | 63.5 |
| OpenCat* [41] | R101+ViT-16 | 32.68 | 28.42 | 33.75 | 61.9 | 66.3 |
| RLIPv2 [37] | R50 | 33.32 | 27.01 | 35.21 | 63.0 | 65.1 |
| DEFR [16] | ViT-16 | 32.35 | 33.45 | - | - | - |
| UPT [39] | R101 | 32.62 | 28.62 | 33.81 | 61.3 | 67.1 |
| ViPLO [28] | ViT-32 | 34.95 | 33.83 | 35.28 | 61.0 | 66.6 |
| ADA-CM [18] | R50+ViT-16 | 33.80 | 31.72 | 34.42 | 56.1 | 61.5 |
| CLIP4HOI [25] | R50+ViT-16 | 35.33 | **33.95** | 35.74 | - | 66.3 |
| *CaM-LQ$_s$* | R50+ViT-16 | **35.67** | 31.90 | **36.80** | 66.4 | 69.8 |
| QAHOI | Swin-L | 35.78 | 29.80 | 37.56 | - | - |
| DiffHOI-L [36] | Swin-L | 40.63 | 38.10 | 41.38 | 63.9 | 65.0 |
| PViC [40] | Swin-L | 44.32 | 44.61 | 44.24 | 64.1 | 70.2 |
| *CaM-LQ$_l$* | Swin-L+ViT-16 | 44.03 | **44.71** | 43.83 | **67.2** | **72.0** |

Table 3: Performance comparison in fully-supervised setting on the HICO-DET and V-COCO datasets. For a fair comparison, we report the results of two-stage methods using an object detector fine-tuned on the training dataset. '*' means using extra data for pre-training. Note that the results of HICO-DET are based on the Default setting.

*5.3.2 Fully-supervised HOI detection.* A robust open-vocabulary HOI model should exhibit strong generalizability while maintaining excellent performance in a closed-set setting. To evaluate this, we conduct experiments in a fully supervised setting. As shown in Table 3, our models with small backbone outperforms all previous methods on the full setting and obtain competitive performance compared to delicately designed models with Swin-L backbone.

## 5.4 Ablation Study

In this section, we conduct ablation experiments on the UC setting of the HICO-DET dataset, using Swin-L as the default backbone unless otherwise specified.

*5.4.1 Effectiveness of Network Architecture.* Firstly, we conducted ablation experiments on the network components as in Table 4.

| Method | Full | Seen | Unseen |
|---|---|---|---|
| Base | 23.55 | 23.62 | 23.31 |
| $+Enc_{sp}$ | 24.64 | 24.94 | 23.46 |
| $+Dec_{sp}$ | 25.29 | 25.76 | 23.41 |
| $+Enc_{sp}+Dec_{sp}$ | 28.38 | 28.51 | 27.89 |
| $+Enc_{sp}+Dec_{sp}+$CaCLIP | 38.00 | 38.90 | 34.39 |

Table 4: Effectiveness of our architecture on Ov-HOI detection. "+sp" means adding spatial embedding into the specific module.

| Method | backbone | Full | Seen | Unseen |
|---|---|---|---|---|
| CLIP | R50 | 30.84 | 30.50 | 32.21 |
| CLIP | ViT-16 | 31.52 | 31.06 | 33.35 |
| RegionCLIP | R50 | 34.44 | 34.50 | 34.19 |
| CaCLIP | R50 | 37.47 | 38.45 | 33.54 |
| CaCLIP | ViT-16 | 38.00 | 38.90 | 34.39 |

Table 5: The performance between different CLIP variants under UC task on HICO-DET.

We defined a base network, i.e., without the addition of any spatial prior information in the encoder or decoder and V&L Teacher. By introducing spatial priors to the queries, the full performance improves by 1.09 points, validating the effectiveness of spatial priors in shaping the features adapted to HOI detection. We further investigate the role of spatial information in the cross-attention mechanism. It can be observed that the full score increases by 1.74 points. Notably, adding the spatial embeddings in the encoder and decoder simultaneously can contribute more mAP gain of 4.83, demonstrating the efficacy of the spatial information.

Finally, we perform ablation on the CaCLIP structure. With the guidance of V&L Teacher, our model exhibits a substantial improvement of 9.62 points in the performance, affirming the significance of our calibrated CLIP for open-vocabulary HOI detection task.

*5.4.2 Noise Introduced by V&L Models.* To investigate the potential noise introduced by V&L models in HOI tasks, we conduct alignment ability experiments on V&L models. RegionCLIP [42] is designed to train a region-level image-text alignment network, with a similarity to our HOI task. Figure 1 illustrates the comparative results of the three models on RTM, UC, UV and UO. RTM denotes region-text matching tasks where we compute the classification precision of region embeddings. It can be observed that RegionCLIP's performance is slightly better than CLIP. However, our CaCLIP significantly outperforms the other two models in HOI classification tasks and open vocabulary tasks, demonstrating the presence of noise in V&L models.

Besides, we also compare different CLIP variants and their calibrated version under UC tasks on the HICO-DET, adding RegionCLIP as well, referring to Table 5. It is noteworthy that we use the weights from pre-trained Region-CLIP to initialize CLIP. Experimental results demonstrate that using ViT-B/16 as the backbone achieves the best performance after fine-tuning. Furthermore, it surpasses Region-CLIP by a large margin of 3.56 points under UC

| Method | Backbone | $AP^{S1}_{role}$ | | | $AP^{S2}_{role}$ | | |
|---|---|---|---|---|---|---|---|
| | | Full | Seen | Unseen | Full | Seen | Unseen |
| CLIP | R50 | 41.49 | 48.62 | 22.21 | 45.92 | 54.66 | 27.04 |
| CLIP | ViT-16 | 42.10 | 49.32 | 22.53 | 48.59 | 55.45 | 27.44 |
| RegionCLIP | R50 | 43.85 | 51.38 | **23.47** | 48.53 | 57.76 | 28.58 |
| CaCLIP | R50 | 44.14 | 52.46 | 22.40 | 48.96 | 63.39 | 27.58 |
| CaCLIP | ViT-16 | **45.98** | **54.65** | 23.33 | **51.00** | **66.03** | **28.73** |

Table 6: The performance between different CLIP variants under the UV task on V-COCO.

| $\lambda_{clip}$ | Full | Seen | Unseen |
|---|---|---|---|
| 1 | 30.40 | 30.45 | 30.18 |
| 100 | 31.36 | 31.50 | 30.84 |
| 200 | 34.52 | 34.54 | 34.47 |
| 300 | 36.28 | 36.84 | 34.02 |
| 400 | 38.00 | 38.90 | 34.39 |
| 700 | 37.23 | 38.03 | 34.03 |

Table 7: The performance with different CLIP loss weight $\lambda_{clip}$ under UC settings.

setting. Plus, we extend our analysis to the V-COCO dataset to provide additional evidence of misalignment in large models regarding HOI visual-semantic information. We provide, for the first time, an Ov-HOI metric on V-COCO and conduct experiments under the UV setting. Specifically, we select 10 out of 24 interaction categories as novel classes, unseen during training, while the model is required to predict both base and novel classes during inference. The experimental results are presented in Table 6. Our CaM-LQ the best performance in both scenarios of $AP^{S1}_{role}$ and $AP^{S2}_{role}$, demonstrating the existence of noise in V&L models and the necessity of calibration and the superiority of our methods in handling Ov-HOI detection.

*5.4.3 Eliminating Noise for CLIP.* To visually demonstrate the improvement in the performance of the refined V&L models on Ov-HOI tasks, we conduct pre-training for varying numbers of epochs. Subsequently, we apply the obtained CaCLIP model to the UC task on the HICO-DET dataset and UV task on the V-COCO, with the results presented in Figure 3 and Figure 4. As the refinement process progresses, it is evident that the performance of the HOI detector utilizing the V&L model gradually improves and hit the peak when training with about 20 epochs and 27 epoch for HICO-DET and V-COCO respectively. This affirms that eliminating noise from V&L models facilitates progress in HOI tasks. Finally, we choose the checkpoints of 20 and 27 epochs as our CLIP models for HICO-DET and V-COCO dataset respectively to deal with all Ov-HOI tasks mentioned above.

*5.4.4 CLIP Loss Weight.* Although we fine-tune CLIP during the pre-training phase, the quality of soft labels is still not as robust as their corresponding hard labels, due to limitations in data scale and significant noise in CLIP's training data. However, soft labels can introduce novel HOI features, which is crucial for Ov-HOI detection tasks. We believe that different loss weights guide the model's attention to different extents for soft labels $\mathcal{L}_{soft}$ and hard labels $\mathcal{L}_{hard}$. We compare the network performance under different loss weight $\lambda_{clip}$ for CLIP, as shown in Table 7. CLIP achieves 30.40 full

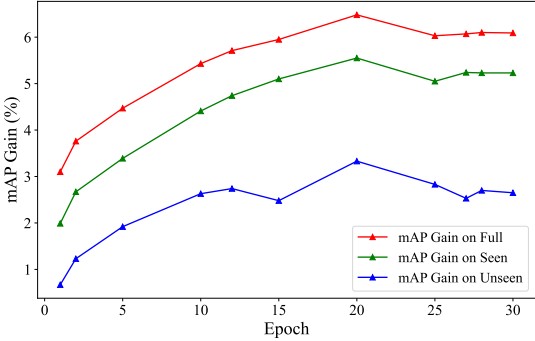

**Figure 3: mAP gain with the process of calibrating CLIP under UC task on HICO-DET.**

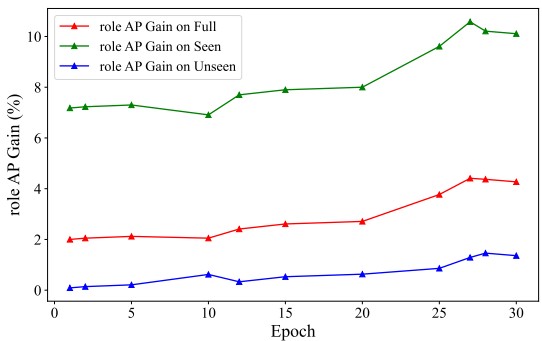

**Figure 4: $AP_{role}^{S_1}$ gain with the process of calibrating CLIP under UV task on V-COCO.**

mAP when the weight is set to 1 and the score grows continuously as loss weight increases, finally hitting the peak at 38.00 mAP as $\lambda_{clip}$ equals to 400. In the case of 700 points, the performance degrades a little bit. Experimental results indicate that under a loss weight of 400, the model can inject the most beneficial open-vocabulary information to aid learning while ensuring effective guidance from hard labels.

*5.4.5 Logits Distillation vs Embedding distillation.* To exploit general knowledge from V&L models, we employ the logits distillation instead of embedding distillation. We conduct preliminary experiments to compare the performance of the two distillation methods in Ov-HOI detection. For logits distillation, we utilize the approach described Sec. 4.4. Regarding embedding distillation, we use the way of GEN-VLKT [20]. Specifically, for an image, we apply CLIP to encode the entire image $\mathcal{I}$ and supervise the average of the query features of the decoder's last layer, i.e., $Q_N \in R^{N_{pair} \times C_r}$, calculating the L1 loss $\mathcal{L}_{mimic}$ as follows:

$$\mathcal{L}_{mimic} = |Adapter_{img}(CLIP_{img}(\mathcal{I})) - \frac{1}{N_{pair}} \sum_{i=1}^{N_{pair}} q_i| \quad (12)$$

where $Adapter_{img}$, $CLIP_{img}$, $N_{pair}$ denote CLIP image adapter, CLIP image encoder and number of feasible pairs, and $q_i \in Q_N$ is the query feature of each pair.

| Method | Full | Seen | Unseen |
|---|---|---|---|
| Logits Distillation | 38.00 | 38.90 | 34.39 |
| Embedding Distillation | 32.25 (-5.75) | 33.06 (-5.84) | 29.62 (-4.77) |

**Table 8: The performance with different distillation schemes: embedding distillation and logits distillation.**

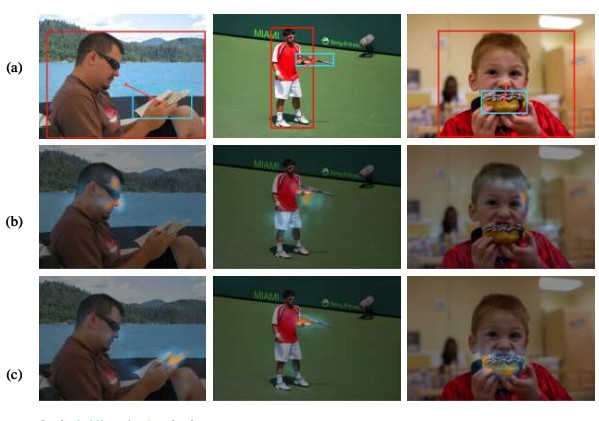

Pred: <hold/open/read•••, book>    Pred: <hold/carry•••, tennis racket>    Pred: <eat/hold/pick_up•••, donut>

**Figure 5: Qualitative results of our CaM-LQ. (a) localization results. (b) Attention maps of the decoder without spatial priors. (c) Attention maps of our full model with spatial priors. Green: correctly detected base category. Orange: correctly detected novel category.**

## 5.5 Qualitative Analysis

Figure 5 presents some visualization results of CaM-LQ. We showcase the localization map and average attention maps from the pairwise decoder, demonstrating CaM-LQ's accurate object localization. By incorporating spatial priors, our attention maps better capture the interaction points between human-object pairs. In terms of category predictions, CaM-LQ can forecast novel categories not present in the training set and even predict reasonable interaction predicates not included in the data labels. This highlights the robust Ov-HOI detection capability of our CaM-LQ.

## 6 CONCLUSION

In this work, we propose an open-vocabulary human-object interaction detection network, CaM-LQ. We identify potential noise issues in the V&L model that negatively influence the HOI task. To this end, we propose a two step open-vocabulary HOI detection model. Firstly, we suppress the intrinsic noise in the CLIP by calibrating the visual-language space of CLIP with HOI priors. Secondly, we inject fine-grained supervision as soft labels deriving from our calibrated CLIP and leverage spatial priors to enhance the target detector's query and assist in extracting feature map information for pairwise queries. Our approach achieves SoTA results on the Ov-HOI setting on the V-COCO and HICO-DET datasets across a wide range of evaluation metrics. For future work, we will focus on integrating the knowledge from large language models (LLM) to provide more explicit cues for CLIP. This endeavor aims to further optimize Ov-HOI tasks, contributing to their enhancement.

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
