# OpenReview forum: "Towards Open-vocabulary HOI Detection with Calibrated Vision-language Models and Locality-aware Queries"
_acmmm.org/ACMMM/2024/Conference — MM2024 Poster_

### Official Review · Reviewer_13nK · 2024-04-28

**Rating:** 4
**Confidence:** 1

**Summary:**

The paper presents a novel two-step framework named CaM-LQ for open-vocabulary human-object interaction (HOI) detection. The framework aims to identify base and novel categories of human-object interactions during training, addressing the limitations of existing methods that struggle with fine-grained detection and are prone to overfitting. CaM-LQ calibrates visual-language models like CLIP with HOI priors and employs locality-aware queries to predict interactions, demonstrating superior performance over state-of-the-art methods on mainstream datasets. The approach involves fine-tuning CLIP to align it with HOI knowledge, using the calibrated CLIP to guide an open-vocabulary HOI detection network through a logit distillation mechanism.

**Strengths:**

+ The code is released anonymously.

+ The proposed model surpasses state-of-the-art approaches.

+ The experiments are sufficient.

+ The method of calibrating the CLIP using the HOI priors seems to be novel.

**Limitations:**

- The latex format is not up to date.

- The paper needs to be proofread. For example, "mode" --> "model" in L147, "denote" --> "denotes" in L230, formulas are not standardized (Eq. 9).

- The grammar used in the paper is confusing. "defiend" in L720, and "improves" in L723, "investigate" in L725.

**Suitability:**

2

---

### Official Review · Reviewer_ZFsf · 2024-05-03

**Rating:** 4
**Confidence:** 2

**Summary:**

The paper presents a novel framework named CaM-LQ (Calibrating visual-language Models with Locality-aware Queries) for open-vocabulary human-object interaction (HOI) detection. It aims to detect both known (base) and unknown (novel) categories of human-object interactions using only base categories for training. Extensive experiments demonstrate that CaM-LQ achieves state-of-the-art performance on standard datasets like HICO-DET and V-COCO across various open-vocabulary settings. In summary, the paper proposes a calibrated vision-language model with locality-aware queries to address the challenge of open-vocabulary HOI detection, demonstrating significant improvements over prior arts. The core innovation lies in refining the visual-semantic space of CLIP for HOI task and leveraging spatial priors to model interactions.

**Strengths:**

The paper detailing the CaM-LQ framework for open-vocabulary human-object interaction detection exhibits several strengths:
1. The authors demonstrate through extensive experiments that CaM-LQ outperforms state-of-the-art methods on multiple metrics across mainstream datasets like HICO-DET and V-COCO. This shows the empirical strength and effectiveness of the proposed model.
2. The authors conduct detailed ablation studies to understand the contribution of different components of the model, which helps in validating the design choices and the effectiveness of the proposed framework.
3. The paper presents a technically sound approach with a clear explanation of the methodology, including the calibration of CLIP, the use of spatial embeddings, and the locality-aware interaction decoder.

**Limitations:**

The method proposed in this paper is commendable. However, it would be beneficial to present a more comprehensive analysis by including additional cases where the model's performance was suboptimal. Extending the discussion to cover a wider array of scenarios, particularly those that highlight the limitations or areas for improvement of the proposed method, would greatly enhance the academic rigor and practical applicability of the research. This could involve detailing specific instances where the model encountered difficulties in accurately predicting human-object interactions, or where the calibration of the vision-language model did not yield the expected results. By doing so, the authors can provide a more nuanced understanding of the model's robustness, its susceptibility to certain types of errors, and offer insights into potential avenues for future research and development.

**Suitability:**

3

---

### Official Review · Reviewer_59ar · 2024-05-24

**Rating:** 4
**Confidence:** 4

**Summary:**

The authors argue that existing methods leveraging knowledge distilled from CLIP (Contrastive Language-Image Pre-training) struggle with fine-grained HOI detection and are prone to overfitting on spatial features of base categories. Moreover, they highlight the issue of misalignment between HOI visual and language knowledge due to noise in pretrained models like CLIP. To address these challenges, the CaM-LQ framework first calibrates CLIP using HOI priors by training two parallel adapters, which helps suppress the intrinsic noise and aligns the visual and text information at a finer granularity. The second step involves using a pre-trained object detector to identify objects and construct pairwise queries for potential human-object pairs. These queries are then refined using spatial embeddings and fed into a locality-aware decoder to predict actions in each HOI triplet.

**Strengths:**

1. The CaM-LQ framework introduces a novel two-step approach to open-vocabulary HOI detection, which is an innovative solution to a challenging problem in computer vision. The idea of calibrating vision-language models like CLIP with HOI priors and using locality-aware queries for fine-grained interaction detection is a fresh perspective in the field.
2. The use of adapters to fine-tune CLIP and the logit distillation mechanism are correct applications of contemporary machine learning techniques.
3. The paper provides a comprehensive evaluation of the CaM-LQ model on mainstream datasets like HICO-DET and V-COCO. The comparison with state-of-the-art methods across multiple metrics demonstrates the effectiveness of the proposed approach.

**Limitations:**

1. While the paper compares the proposed CaM-LQ model with CLIP and its variants, it may benefit from a broader comparison with other SOTA VLMs that have been released around the same time or after. This would provide a more comprehensive understanding of where CaM-LQ stands in the broader context of VLM advancements.
2. The paper reports strong performance on the HICO-DET and V-COCO datasets. However, it is not clear how well the model generalizes to other datasets or real-world scenarios. There might be a risk of overfitting to the specific characteristics of the datasets used for evaluation.
3. While the paper demonstrates the model's effectiveness, there is limited discussion on the interpretability of the model's predictions. Understanding how the model makes its predictions, especially in an open-vocabulary setting, is crucial for trust and adoption in practical applications.
4. The paper relies on automated metrics for evaluation. Including human subject evaluation, such as user studies, could provide additional insights into the model's performance from a practical standpoint.

**Suitability:**

3

---

### Official Review · Reviewer_VG54 · 2024-05-28

**Rating:** 3
**Confidence:** 3

**Summary:**

The paper proposes a two-step framework called CaM-LQ for open-vocabulary human-object interaction (HOI) detection. This approach incorporates fine-grained HOI supervision from a calibrated CLIP model into the HOI decoder and injects spatial priors into HOI queries. Extensive experiments and ablation studies demonstrate the effectiveness of this method.

**Strengths:**

1.	The 2-step framework is intuative and tenichally sound for the HOI deteciton task.
2.	The method achieves impressive performance on various benckmarks.
3.	Ablation study confirm the effectivness of various design choices.

**Limitations:**

1.	What is the difference between the claimed open-vocabulary setting and previous zero-shot setting in the HOI detection community?
2.	What’s the method’s performance on the open-vocabulary HOI dataset: SWIG-HOI [1]?
3.	The calibration procedure may lead to overfitting to the seen classes. I find that in Table 5 and 6, the improvement brought by CaCLIP is mainly on the seen split. I also notice that when utilizing R50 backbone, CaCLIP shows inferior perforamnce on the unseen split compared with RegionCLIP, which supports my hypothesis. Please provide a detailed analysis of the phenomenon.
4.	I doubt the effectiveness of the designed spatial prior injection. Please provide more ablation study on this design.
5.	What’s the total training time of the proposed method? Please compare it with previous methods.
6.	Can the proposed training paradigm be applied to different VLMs, such BLIP or BLIP2?
7.	Writing issue: the equation (8) should be center-aligned.


[1] Wang, Suchen, et al. "Discovering human interactions with large-vocabulary objects via query and multi-scale detection." ICCV 2021

**Suitability:**

2

---

### Meta-Review · Area_Chair_s7FJ · 2024-07-07

**Recommendation:** Accept (Poster)
**Confidence:** 5

**Metareview:**

This paper proposes a two-step framework for open-vocabulary human-object interaction (HOI) detection. Initially, the paper received 3 borderline accept and 1 borderline reject (Reviewer VG54).  Reviewer VG54 used to have concern on the definition of the open-vocabulary HOI task, the performance on a specific dataset, the effectiveness of some component. During rebuttal, the authors addressed these concerns by providing clarification on the task, the required experimental results, as well as the additional ablation study. All the reviewers reached consensus that this paper is acceptable.